# Corporate Sustainable Development, Corporate Environmental Performance and Cost of Debt

**Haiyan Sun [1], Guangyang Wang [1], Junwei Bai [1], Jianfei Shen [2,*], Xinyuan Zheng [2], Erli Dan [2], Feiyu Chen [2] and Ludan Zhang [2]**

[1]  State Grid Materials Co., Ltd., Beijing 100120, China
[2]  Economics and Management Department, North China Electric Power University, Beijing 102206, China
*   Correspondence: xinyzheng@ncepu.edu.cn

**Abstract:** High environmental performance of enterprises may reduce financing costs, while good environmental performance can promote sustainable development of enterprises. Therefore, this paper examines the impact mechanism of China's corporate environmental performance on financing costs, and whether corporate sustainable development plays a regulatory role in the research of heavy pollution industries. This study is conducted through the Breusch and Pagan Lagrange multiplier test for random effects and the Hausman test to determine whether to adopt Fixed-effects regression or Random-effects GLS regression as an estimation method to control individual effects and endogenous problems brought by time. By collecting the samples of listed companies in China from 2010 to 2021, the empirical results show that corporate environmental performance is negatively related to financing costs. Sustainable development, as a moderator variable, is negatively related to financing costs and has weakened the inhibition of corporate environmental performance on financing costs. Although the existing literature shows that environmental performance will lead to changes in debt costs, this study has made contributions to the literature by revealing the sustainable development mechanism in the relationship between corporate environmental performance and financing costs and has verified that sustainable development is one of the important factors affecting financing costs.

**Keywords:** corporate sustainable development; corporate environmental performance; cost of debt; costs of capital; sustainability; China

## 1. Introduction

With the diversification of economic development, in order to consolidate and further improve market share and competitiveness, enterprises need to solve the problem of insufficient supply of their own funds, so external financing is increasingly important. Although China's capital market started late, it has developed rapidly. While promoting the sustainable development of the national economy, it has greatly eased the pressure on corporate financial leverage [1]. However, the external financing of listed companies in China is still dominated by debt financing, especially bank loans, and the proportion of equity financing is relatively low. The relatively high cost of debt capital not only limits the company's capacity and R&D scale expansion through borrowing but also has a negative impact on the enterprise's operating performance and capital situation, which greatly aggravates the potential financial distress risk faced by the enterprise [2]. Therefore, how to reduce the cost of debt to improve the financial performance and financing burden of enterprises, and even positively affect the subsequent financing and profitability, is still the focus of academic circles and research topics.

Brundtland first proposed the concept of "sustainable development", which is defined as "development that not only meets the needs of contemporary people, but also does not harm the ability of future generations to meet their needs" [3]. It requires that the

triple bottom lines of long-term economic prosperity, social equity, and environmental responsibility be included in the business practice and management of enterprises [4]. Brundtland's definition of the concept of sustainable development has had a profound impact on scholars' subsequent research [5]. For today's society, the core of sustainable development has shifted from constant economic growth to high-quality economic development. As the main body of social economic operation, enterprises play an important role in the process of high-quality economic development. Enterprises with large scale and influence have all launched sustainable development plans, while SMEs limited by resources and funds have lagged in sustainable development [6].

In recent years, the management of some enterprises, in order to promote the stock price and deliberately pursue short-term high-speed growth, often ignored the potential crisis of sustainable development of enterprises, resulting in a shortened enterprise life cycle. The sustainable development of an enterprise should not only consider the maximization of short-term shareholder wealth but also consider the importance of capital demand to the sustainable operation and investment of the enterprise from a strategic perspective [7]. At the same time, according to the information asymmetry theory, enterprises should fully disclose their sustainable development information to their financing providers, such as "capital markets, banks and suppliers", especially for financial liabilities. Banks and other financial institutions, out of prudence, are more willing to lend to enterprises with strong sustainable development ability in due diligence. Therefore, it is meaningful to study the internal relationship between the sustainable development capability of enterprises and their access to financing.

At present, the global environmental situation shows that the problem of environmental degradation is becoming more and more serious, and all sectors of society pay more and more attention to the environmental behavior of enterprises. At the same time, with the further deepening of environmental protection education, the public began to actively supervise and report the environmental violations of enterprises. Real-time environmental data released by some environmental research institutions can also, in a timely manner, urge enterprises to correct environmental pollution violations. The level of enterprise environmental performance has become one of the important criteria for investors to measure enterprises. Investors will be more inclined to choose enterprises with better environmental behavior, thus effectively reducing the cost of equity capital of enterprises. In this context, enterprises in various countries, especially those in heavily polluting industries, pay more attention to the environmental impact brought by the implementation of the production process and the implementation of corresponding undertaking measures. As an important aspect and institutional guarantee for enterprises to achieve environmental sustainability, the environmental performance of heavily polluting enterprises has been highly concerning to the government, the public, and other stakeholders [8]. In fact, good corporate environmental performance is conducive to protecting the interests and image of enterprises. According to the signaling theory, enterprises can send positive signals to stakeholders through good environmental performance. Sustainable development capability is also a part of enterprise values and business methods. In the process of putting its sustainable development values into practice, enterprises also need to disclose information to stakeholders as comprehensively as possible. The disclosure should include not only the economic performance of the enterprise but also the environmental performance [6]. Improving environmental performance has been widely recognized as a key action to promote environmental sustainability. According to environmental sustainability regulations, enterprises may be eliminated if they do not actively transform and upgrade [9]. Therefore, it is necessary to explore the relationship between environmental performance, sustainable development, and financing costs from multiple perspectives.

The existing literature focuses on the economic effects caused by environmental performance, and its economic effects pay more attention to enterprise innovation [9,10]. From the perspective of corporate financing costs, this paper discusses the impact of environ-

mental performance on corporate financing constraints, which enriches the research on the economic consequences of environmental performance. At the same time, the relationship between sustainable development and corporate financing constraints has always been an important topic of concern for scholars, but existing research only focuses on the relationship between debt heterogeneity and property rights, and few studies discuss the relationship between sustainable development and financing costs [11]. This paper studies the influencing factors of financing cost constraints from the perspective of sustainable development. It expands the research on influencing factors of corporate financing constraints. To sum up, the important research questions of this paper are as follows: 1. Can corporate environmental performance reduce financing costs? 2. Can sustainable development reduce financing costs? 3. Is there a synergy between sustainable development and environmental performance on corporate financing costs?

This study has two contributions to the literature. First of all, at present, some studies have investigated the impact of environmental information disclosure and other environmental factors on the cost of debt [12,13], but the research on how environmental performance affects corporate debt is limited. Our empirical results support that good environmental performance can reduce the cost of debt. From the company's perspective, this analysis helps to determine the effectiveness and economic significance of the environmental performance. Secondly, other researchers did not link the sustainable development ability of enterprises with the cost of debt, but only focused on the impact of other enterprise factors such as the financial performance of enterprises on the cost of debt [1,14]. In this paper, environmental performance is used as a moderator variable, so that stakeholders can better understand the importance of environmental performance. This paper determines for the first time the regulatory role of corporate environmental performance in the relationship between corporate sustainable development capability and debt cost. The company's environmental management, environmental performance monitoring and enterprise sustainable development all benefit significantly from the results of this study.

The rest of this paper is organized as follows: Section 2 reviews the literature and develops our hypotheses. Section 3 conducts the collected data and illustrates the methodology. We present the empirical findings in Section 4 and conclude our paper and provide discussions in Section 5.

## 2. Literature Review and Hypothesis Development

The financing cost of an enterprise refers to various expenses incurred by the enterprise in the process of raising its own funds. It is the product of the separation of capital ownership and user rights. From the perspective of the market, enterprise financing is one of the acts of market transactions, and transaction costs are an inevitable element [15]. These expenses are what capital users must pay in order to obtain the right to use funds. These include but are not limited to agency fees, registration fees, etc. related to the entrusted issuance of bonds and stocks, and the most basic handling fees paid for bank loans. Therefore, in short, the essence of financing costs is the remuneration paid by the users of funds to the owners of funds.

In the actual operation of the company, many factors will affect the financing cost of the enterprise. As early as 1973, there had been relevant empirical studies that confirmed that default risk would lead to an increase in debt financing costs [16,17]. The impact of environmental performance on corporate debt financing costs is related to green credit policies and pricing methods of various debt funds [2]. The Green Credit Guidelines suggest that banks dynamically assess and classify customers' environmental risks and take differentiated risk management measures in loan pricing. The cost of debt funds is the compensation that creditors get by transferring funds to bear risks. The higher the risk of default, the higher the debt financing cost. Enterprise environmental performance is related to enterprise environmental risk [18]. The potential environmental risk of an enterprise

will have an impact on the company's value, thereby increasing or reducing the company's debt default risk and affecting the company's financing costs [13].

## 2.1. Corporate Environmental Performance and Financing Cost

The environmental performance reflects the environmental risk of the enterprise, which will affect the business performance and enterprise value to a certain extent, thus indirectly affecting debt financing through the default risk of the enterprise. According to the literature, scholars have reached a consensus that environmental performance has a positive impact on financial performance [19,20]. Scholars also have different opinions on the specific impact of environmental performance on companies. Some believe that environmental performance is related to the effective use of resources for the overall improvement of organizational efficiency [21], while others believe that environmental performance will improve corporate image and reputation, which may help enterprises gain benefits in the labor, commodity and capital markets [22]. Other scholars believe that investment in environmental performance will reduce the operational risk of enterprises, and the reduction of operational risk can make it easier for enterprises to obtain funds and even reduce financing costs [23,24]. Thompson et al. through questionnaires and semi-structured interviews with the Bank of England, which granted corporate loans, investigated that the public's attention to environmental information has become the motivation for banks to pay attention to corporate environmental information, and banks will use the environmental performance information in the corporate annual report when making loan decisions [25]. Du et al. used content analysis to prove that the company's environmental performance had a significant negative impact on the cost of debt [26].

On the verification of specific industries, some scholars, through measuring and judging the environmental risk of debt markets such as real estate, have concluded that environmental performance has an impact on the debt market of real estate, and even environmental risk itself has its definite value in the market [27]. Further, in Europe, which focuses on the environment, some scholars have confirmed that the generally positive impact of carbon emissions on financial debt that was induced by the role of emissions as an indicator of activity was mitigated by firms' carbon environmental performance. Therefore, better environmental performance can lead to better carbon emission performance, while excellent carbon emission performance enables industrial companies to obtain more long-term financial debt, feedback on their environmental investment, and provide funds for their related environmental investment.

Now that environmental problems are increasingly prominent, the academic community is increasingly focusing on the relevant impact of the environment on the company. Environmental performance is one aspect of the company's management, and financing cost is the external embodiment of the company's management. The relationship between the two still needs to be further confirmed and studied. Therefore, hypothesis 1 is proposed in this study:

**H1.** *Corporate environmental performance lowers the cost of debt financing.*

## 2.2. Corporate Sustainability and Financing Cost

The ability of sustainable development of an enterprise refers to the ability of an enterprise to achieve its business objectives while maintaining its advantages, making profits, and growing steadily in the future for its long-term survival and development. However, due to the complexity and variability of the company's business process, there is no unified definition and measurement standard for the sustainable development capability of enterprises in academic circles at home and abroad. Some scholars have proposed that the sustainable development capability possessed by enterprises may include the awareness and ability of independent innovation, the efficiency and ability of resource allocation, and the enterprise's own business ability [11]. There are also many classifications for the standard division of sustainable development capability. From the corporate governance level, it can be divided into strategic development capability, production and operation

capability, etc., and from the financial management level, it can be divided into operation capability, profitability, solvency, growth capability, etc. From the perspective of relatively simple and single data, some scholars also measured the sustainable development ability of enterprises by calculating the net profit growth level of enterprises in the last three years [28]. These indicators, which can be used to indicate sustainable development capacity are all reference information in the financing process and affect the company's financing situation.

Some studies show that the scale of corporate assets and profits, organizational structure, asset liability maturity structure, financial leverage, discretionary cash flow, and other factors significantly affect the cost of corporate debt [1,29,30]. Based on the research of scholars on financing costs, previous studies have focused more on the role and impact of enterprises' own behavior on financing costs. Corporate finance, development, and operational capabilities will all have an impact on financing costs. As an important part of sustainable development capabilities, corporate governance capabilities and enterprise development capabilities will have a certain effect on financing costs to some extent. Some scholars pointed out that the growth ability of enterprises is negatively related to financing costs [31,32]. In terms of development ability, under the conditions of a chaotic market environment and imperfect system, the effect of improving financial performance and other development abilities to reduce financing costs will be more obvious [8].

Although the academic research on enterprise financing costs is deep and extensive, the research directly linking the sustainable development ability of enterprises with financing cost is still in the exploration stage. Therefore, Hypotheses 2 and 3 are proposed in this study, and Figure 1 shows the research ideas of this paper.

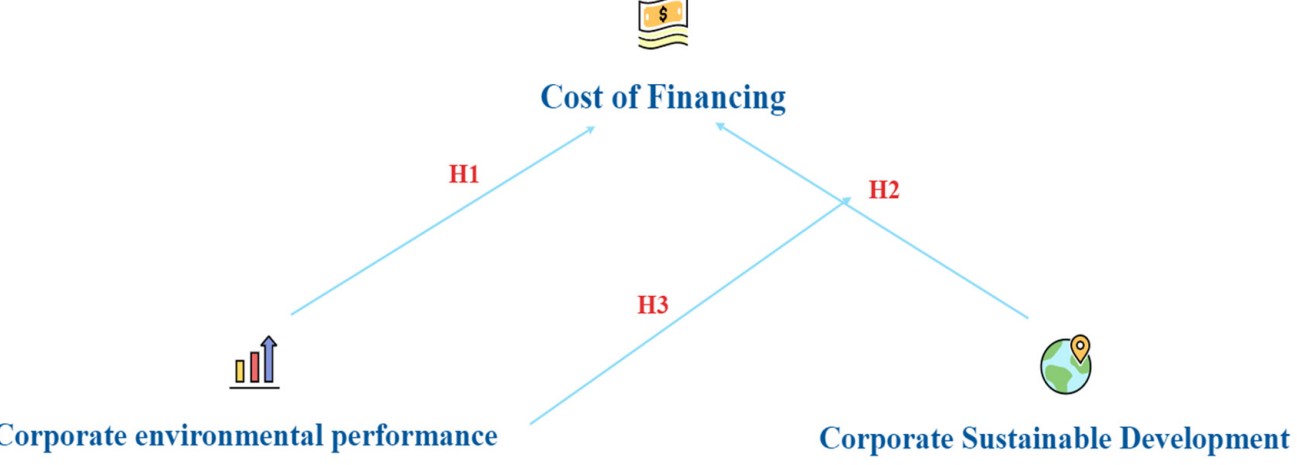

**Figure 1.** Illustrates the interactions predicted in the paper and which are to be tested in empirical analyses.

**H2.** *Corporate sustainable development negatively affects the cost of debt financing.*

**H3.** *Corporate sustainable development weakens the relationship between corporate environmental performance and debt costs.*

## 3. Methodology

### 3.1. Sample Selection and Data Sources

Our sample consists of A-share listed companies in the heavily polluting industries in China over the period 2010–2021. According to the definition in the Guidelines for Environmental Information Disclosure of Listed Companies (Draft for Comments) issued by the Ministry of Ecology and Environment, formerly the Ministry of Environmental Protection of China (MEP), in 2010 (Draft Guidelines for EID 2010, hereafter), heavily polluting industries in our study include thermal power, iron and steel, cement, electrolytic aluminum, coal,

metallurgy, chemical, petrochemical, building materials, papermaking, brewing, pharmacy, fermentation, textile, tanning, and mining. We obtained data on carbon disclosure from annual reports and CSR reports posted on publicly listed companies' official websites. The annual reports, the CSMAR database, and the RESSET database are the sources of information on control variables. The 99% quantiles were abbreviated and normalized for the main variables, and explanatory and moderator variables were centered before constructing interaction terms. Applying the abovementioned filters yields our final panel dataset consisting of 786 firm-year observations.

*3.2. Variables Definitions*

The dependent variable in the model is the debt of cost (COST). At present, scholars have no consistent measurement standard for debt costs [33–36]. Considering the data availability and China's actual condition, this paper uses the method of Pittman and Fortin, which is the ratio of interest expenses and interest-bearing debt to measure the cost of debt financing [37].

The main representative point of the sustainable growth theory comes from Professor Higgins, who proposed that the sustainable growth rate is the maximum sales that enterprises can achieve without issuing new shares or changing operating efficiency and financial policies [38]. The growth rate is a more accurate measure of a company's sustainability. Therefore, sustainable growth rate (SUS) was chosen as the explanatory variable for corporate sustainability in this study.

In this paper, the unit operating income pollution charge is used as the proxy variable of environmental performance (CEP) [39]. The lower the pollution charge per unit operating income of an enterprise, the better the enterprise's control of pollutant emissions, and the higher the enterprise's level of environmental performance.

Following Wu et al. among others, this paper includes a set of control variables to control the possible effects of corporate characteristics, financial status, and corporate governance on bank financing [39]. We select the firm size (SIZE), return on total assets (ROA), financial leverage (LEV), fixed asset ratio (FIX), quick ratio (QUICK), CSR participation (CSRR), equity concentration 10 (OC) and nature of equity (OS) as control variables. This study chooses natural logarithm of operating income as proxy variables of company size. Generally, larger companies have more capital accumulation, stronger sustainable development capabilities, higher environmental performance, and lower financing costs [40]. We use return on total assets to represent the company's profitability. The lower the required rate of return, the less dependent the firm is on outside capital to grow reproduction [29]. Consequently, we set ROA as a control variable. The long-term debt-to-total-assets ratio represents the debt risk and the default risk. The long-term debt-to-total-assets ratio is an important factor affecting capital cost. From the perspective of solvency and capital utilization, quick ratio (QUICK) and fixed asset ratio (FIX) are selected as control variables. Companies that release social responsibility reports voluntarily have a better chance of lowering bond financing costs than those that must disclose in compliance with requirements [41]. Consequently, this paper employs the CSR participation of the company as a control variable. Equity concentration is related to corporate governance and corporate decision-making and is an important indicator to measure the status of corporate equity distribution, corporate stability, and corporate structure, and may have an impact on financing costs [42]. This study sets equity concentration as the control variable, and accordingly takes equity concentration 10 (OC) as the control variables. The company's growth is a major constraint on corporate debt financing [43]. The specific descriptions of all the variables involved in this model are shown in Table 1.

**Table 1.** Definition of Variables.

| Variables | Name | Code | Definition |
|---|---|---|---|
| Dependent variable | Cost of debt | COST | Cost of debt. Interest expense/the average of long-term debt and short-term debt |
| Independent variable | Corporate environmental performance | CEP | The unit operating income pollution charge |
| | Corporate sustainable development | SUS | (current net profit/beginning shareholders' equity) × current earnings retention rate × 100% |
| Control variables | Leverage | LEV | Long-term debt/total assets. |
| | Firm size | SIZE | The natural logarithm of total revenue. |
| | Return on assets | ROA | Net income/total assets. |
| | Fixed asset ratio | FIX | Fixed asset ratio |
| | Quick ratio | QUICK | Quick Asset/Current liabilities |
| | Issue CSR reports | CSRR | CSR reporting, coded 1 if a company issues CSR report, code 0 otherwise |
| | Shareholding ratio | OC | Shareholding ratio of top 10 shareholders |
| | State-owned | OS | Dummy variable. Code 1 if State-owned, code 0 otherwise |
| | Year control | YEAR | Year control |
| | Industry control | INDUSTRY | Industry control |

### 3.3. Model Specification

To examine H1, H2, and H3, the following models are adopted.

$$COST_{i,t} = \alpha + \beta_1 CEP_{i,t} + \beta_2 SIZE_{i,t} + \beta_3 FIX_{i,t} + \beta_4 ROA_{i,t} + \beta_5 LEV_{i,t} + \beta_6 QUICK_{i,t} + \beta_7 CSRR_{i,t} + \beta_8 OC_{i,t} + \beta_9 OS_{i,t} + YEAR + INDUSTRY + \varepsilon_{it}$$

$$COST_{i,t} = \alpha + \beta_1 SUS_{i,t} + \beta_2 SIZE_{i,t} + \beta_3 FIX_{i,t} + \beta_4 ROA_{i,t} + \beta_5 LEV_{i,t} + \beta_6 QUICK_{i,t} + \beta_7 CSRR_{i,t} + \beta_8 OC_{i,t} + \beta_9 OS_{i,t} + YEAR + INDUSTRY + \varepsilon_{it}$$

$$COST_{i,t} = \alpha + \beta_1 CEP_{i,t} + \beta_2 SUS_{i,t} + \beta_2 CERP \times SUS_{i,t} + \beta_4 SIZE_{i,t} + \beta_5 FIX_{i,t} + \beta_6 ROA_{i,t} + \beta_7 LEV_{i,t} + \beta_8 QUICK_{i,t} + \beta_9 CSRR_{i,t} + \beta_{10} OC_{i,t} + \beta_{11} OS_{i,t} + YEAR + INDUSTRY + \varepsilon_{it}$$

We employ logit models to model dichotomous dependent variables. The logit model shows better performance than the Probit model since it is more appropriate for modeling log odds. *YEAR* and *INDUSTRY*, respectively, represent the time effect and industry effect of the company. This paper controls these effects in the regression analysis to control time trends and differences in the industry.

## 4. Results

### 4.1. Descriptive Statistics

Table 2 shows the descriptive statistical results of each variable. It can be seen from the table that the average COST is 0.095, the standard deviation is 0.083, the minimum value is 0.019, and the maximum value is 0.650, indicating that there is a large difference in financing levels among enterprises. The average value of CEP is 6894.103, the standard deviation is 26,174.3, the minimum value is 85.302, and the maximum value is 377,096.6, indicating that the level of the environmental performance of enterprises is uneven. China should actively promote the linkage between the sustainable development of enterprises and environmental performance, and also the autonomy of enterprises in formulating sustainable development strategies and participating in environmental protection. The average value of SUS is 1.660, the standard deviation is 141.260, the minimum value is −537.496, and the maximum value is 464.762, indicating that there is a large difference in the level of sustainable development among enterprises. The average value of financial leverage (LEV) is 0.145, the maximum value is 0.765, and the minimum value is 0. It can be seen that the overall financial risk of the heavily polluting industry is moderate, but the difference between enterprises is large; The average value of enterprise size (SIZE) is

22.607, and the standard deviation is 1.437, indicating that the sample enterprises have a good size with a small difference; Profitability (ROA), that is, return on total assets, has an average of 1.642, a maximum of 68.642, and a minimum of −64.157. It can be seen that the overall performance level of heavily polluting industries is low, and the difference between enterprises is significant. The standard deviation of fixed asset ratio (FIX) is 15.881, with a high degree of dispersion and a large difference in asset size between enterprises; The average QUICK ratio is 0.730, indicating that heavily polluted enterprises have strong short-term solvency; The average value of social responsibility report (CSRR) is 0.382, indicating that more enterprises release social responsibility reports and disclose environmental information; The equity concentration (OC) is the sum of the shareholding ratios of the top ten shareholders, with an average of 0.578. The degree of dispersion is high, and the equity of some samples is relatively concentrated; The standard deviation of equity nature (OS) is 0.415, which indicates that both state-owned enterprises and non-state-owned enterprises are involved in the heavily polluting enterprises, and the sample is relatively reasonable. In general, the sample data is not volatile and discrete, with good stability. The sample selection is reasonable and representative.

**Table 2.** Descriptive Statistics.

| Variable | Obs | Mean | Std. Dev. | Min | Max |
|---|---|---|---|---|---|
| COST | 786 | 0.095 | 0.083 | 0.012 | 0.650 |
| CEP | 786 | 6894.103 | 26,174.3 | 85.302 | 377,096.6 |
| SUS | 786 | 1.660 | 141.260 | −537.496 | 464.762 |
| LEV | 786 | 0.145 | 0.119 | 0 | 0.765 |
| SIZE | 786 | 22.607 | 1.437 | 18.327 | 25.616 |
| ROA | 786 | 1.642 | 7.277 | −64.157 | 68.642 |
| FIX | 786 | 40.045 | 15.881 | 0.021 | 85.033 |
| QUICK | 786 | 0.730 | 0.841 | 0.064 | 12.040 |
| CSRR | 786 | 0.382 | 0.486 | 0 | 1 |
| OC | 786 | 0.578 | 0.161 | 0.207 | 0.961 |
| OS | 786 | 0.779 | 0.415 | 0 | 1 |

*4.2. Correlation Coefficients*

Table 3 is the correlation analysis between variables. The correlation coefficient between SUS and COST is −0.394, which is significantly negative at the level of 1%, indicating that the financing cost is decreasing with the improvement of the sustainable development level of enterprises; CEP and COST are significantly negatively correlated, indicating that corporate environmental performance increases while promoting the reduction of financing costs. LEV, SIZE, FIX, CSRR, OC, and OS are negatively related to financing costs; ROA and QUICK are positively correlated with financing costs, both of which are significant at the level of 1%. The values of the model's correlations are shown in Table 3, with pairwise correlations for the independent and control variables both being less than 0.5. As a result, the variables in this sample are without multicollinearity risk.

**Table 3.** Correlation coefficients result.

|  | COST | CEP | SUS | LEV | SIZE | ROA | FIX | QUICK | CSRR | OC | OS |
|---|---|---|---|---|---|---|---|---|---|---|---|
| COST | 1 | | | | | | | | | | |
| CEP | −0.306 *** | 1 | | | | | | | | | |
| SUS | −0.394 *** | 0.185 *** | 1 | | | | | | | | |
| LEV | −0.214 *** | 0.0200 | 0.0370 | 1 | | | | | | | |
| SIZE | −0.0320 | 0.237 *** | 0.0310 | 0.198 *** | 1 | | | | | | |
| ROA | 0.200 *** | −0.073 ** | 0.311 *** | −0.136 *** | 0.090 ** | 1 | | | | | |
| FIX | −0.0570 | −0.170 *** | 0.00700 | 0.281 *** | 0.119 *** | −0.0250 | 1 | | | | |
| QUICK | 0.147 *** | −0.063 * | 0.0100 | −0.167 *** | −0.255 *** | 0.141 *** | −0.295 *** | 1 | | | |
| CSRR | −0.0590 | 0.085 ** | −0.00200 | 0.127 *** | 0.160 *** | −0.0380 | 0.065 * | 0.0100 | 1 | | |
| OC | −0.0160 | 0.0570 | 0.0110 | 0.0480 | 0.081 ** | 0.0190 | 0.0460 | 0.0170 | 0.161 *** | 1 | |
| OS | −0.133 *** | 0.0270 | 0.0170 | 0.240 *** | 0.137 *** | −0.0380 | 0.065 * | 0.0200 | −0.160 | −0.108 *** | 1 |

*** $p < 0.01$, ** $p < 0.05$, * $p < 0.1$.

### 4.3. Regression Analyses

Table 4 provides the logit regression results of the relationship between sustainable development of enterprises, carbon information disclosure, and environmental costs. In three different models, the author estimated the relationship between the sustainable development of enterprises, carbon information disclosure, and environmental costs. The first model (Model 1) shows the relationship between CEP and COST under the control of SUS and other relevant variables.

**Table 4.** Regression result.

|  | **Model 1** | **Model 2** | **Model 3** |
|---|---|---|---|
| VARIABLES | COST | COST | COST |
| CEP | −0.020 *** | | −0.197 *** |
| | (−9.13) | | (−6.84) |
| SUS | | −0.097 *** | −0.169 *** |
| | | (−15.44) | (−11.88) |
| CEP × SUS | | | 0.019 *** |
| | | | (6.40) |
| SIZE | 0.006 *** | 0.001 | 0.004 * |
| | (3.07) | (0.68) | (1.94) |
| FIX | −0.000 | 0.000 | −0.000 |
| | (−1.03) | (0.78) | (−0.62) |
| ROA | 0.001 *** | 0.004 *** | 0.002 *** |
| | (3.73) | (9.68) | (6.18) |
| LEV | −0.110 *** | −0.083 *** | −0.077 *** |
| | (−4.42) | (−3.61) | (−3.51) |
| QUICK | 0.010 *** | 0.010 *** | 0.008 *** |
| | (2.87) | (3.13) | (2.65) |
| CSRR | −0.004 | −0.006 | −0.007 |
| | (−0.71) | (−1.20) | (−1.37) |
| OS | −0.020 *** | −0.019 *** | −0.020 *** |
| | (−2.90) | (−3.00) | (−3.36) |
| OC | −0.004 | −0.010 | 0.001 |
| | (−0.26) | (−0.65) | (0.07) |
| Constant | 0.115 ** | 1.021 *** | 1.761 *** |
| | (2.46) | (13.64) | (12.13) |
| YEAR | YES | YES | YES |
| INDUSTRY | YES | YES | YES |
| Observations | 786 | 786 | 786 |
| R-squared | 0.183 | 0.308 | 0.379 |

*t*-statistics or z-statistics in parentheses; *** $p < 0.01$, ** $p < 0.05$, * $p < 0.1$.

The results of the first model show that corporate environmental performance is negatively correlated with financing costs. In model 1, the coefficient of CEP is 0.020, which indicates that every 1% increase in corporate environmental performance will reduce the financing cost by 0.020 points. Research H1 has been verified, and the results are consistent with previous studies [24,44].

The second and third models (Model 2 and Model 3), respectively, give the relationship between SUS and CEP. In order to further test the impact of CEP on the relationship between enterprise SUS and COST, this study adds the interaction term of CEP and SUS, which is expressed by CEP × SUS. The results of Model 2 and Model 3 show that SUS has a significant negative correlation with COST, and research H2 has been verified. In model 3, the coefficient of CEP is −0.197, the coefficient of SUS is −0.0169, and the coefficient of the interaction item is 0.019, which is significant at the level of 1%. This shows that the improvement of corporate environmental performance has strengthened the inhibition of corporate sustainable development on financing costs. Research H3 has been verified, and the research results have economic significance.

Although the empirical results are significant, there are still limitations. Because the sample size is not large enough, it may not be completely representative. In addition, the impact of the epidemic situation was not considered when selecting data, which may lead to a deviation between the data results and the actual situation.

### 4.4. Robustness Checks

In view of the conclusion that the environmental performance of enterprises in China's heavy pollution industries can reduce the cost of debt financing, this study uses alternative independent variables to test its robustness. That is, the return on equity of the enterprise (ROE) is replaced by the return on equity of the enterprise. The experiment shows that the fixed model is feasible. After correcting heteroscedasticity and autocorrelation, the results show that the more environmental performance disclosure, the lower the debt cost, as shown in Table 5. In addition, sustainable development weakens the inhibition of environmental performance disclosure on debt cost, which is consistent with the original results.

**Table 5.** Robustness result.

| | **Model 1** | **Model 2** | **Model 3** |
|---|---|---|---|
| Variables | COST | COST | COST |
| CEP | −0.020 *** | | −0.197 *** |
| | (−9.13) | | (−6.84) |
| SUS | | −0.097 *** | −0.169 *** |
| | | (−15.45) | (−11.88) |
| CEP × SUS | | | 0.019 *** |
| | | | (6.39) |
| SIZE | 0.006 *** | 0.001 | 0.004 * |
| | (3.06) | (0.67) | (1.93) |
| FIX | −0.000 | 0.000 | −0.000 |
| | (−1.03) | (0.78) | (−0.62) |
| ROA | 0.001 *** | 0.004 *** | 0.002 *** |
| | (3.75) | (9.70) | (6.21) |
| LEV | −0.110 *** | −0.083 *** | −0.077 *** |
| | (−4.42) | (−3.62) | (−3.52) |
| QUICK | 0.010 *** | 0.010 *** | 0.008 *** |
| | (2.86) | (3.12) | (2.64) |
| CSRR | −0.004 | −0.006 | −0.007 |
| | (−0.71) | (−1.21) | (−1.38) |
| OS | −0.020 *** | −0.019 *** | −0.020 *** |
| | (−2.89) | (−2.99) | (−3.35) |
| OC | −0.004 | −0.010 | 0.001 |
| | (−0.26) | (−0.64) | (0.08) |
| Constant | 0.116 ** | 1.023 *** | 1.762 *** |
| | (2.48) | (13.66) | (12.14) |
| YEAR | YES | YES | YES |
| INDUSTRY | YES | YES | YES |
| Observations | 786 | 786 | 786 |
| R-squared | 0.183 | 0.309 | 0.379 |

*t*-statistics or z-statistics in parentheses. *** $p < 0.01$, ** $p < 0.05$, * $p < 0.1$.

### 4.5. Endogeneity Considerations

In the regression analysis, the author needs to consider the endogeneity problem. Therefore, in order to reduce the possible endogeneity and self-selection bias, we adopted the replacement variable analysis method [44]. This paper uses the model of financial

sustainable growth reference to further measure the sustainable development capability of enterprises:

$$CSDC = \frac{NPMS \times RPM \times (1 + Equity\ ratio)}{\frac{1}{TRTA} - NPMS \times RPM \times (1 + Equity\ ratio)}$$

where
   *NPMS* = Net profit margin on sales
   *RPM* = Retained profit margin
   *TRTA* = Turnover rate of total assets

Table 6 shows the results of the correlation between enterprise environmental performance and financing cost by using alternative enterprise sustainable development variables. The environmental performance of the company is significantly negatively correlated with the financing cost, and the SUS is significantly negatively correlated with the COST. The improvement of environmental performance strengthens the inhibition of corporate sustainable development on the financing cost. H1, H2, and H3 are verified, and consistent with the original results.

**Table 6.** Endogeneity result.

| Variables | Model 1 COST | Model 2 COST | Model 3 COST |
|---|---|---|---|
| CEP | −0.020 *** | | −0.012 *** |
| | (−9.13) | | (−6.76) |
| SUSA | | −0.025 *** | −0.097 *** |
| | | (−14.76) | (−20.42) |
| CEP × SUSA | | | 0.014 *** |
| | | | (16.49) |
| SIZE | 0.006 *** | 0.001 | 0.003 * |
| | (3.07) | (0.59) | (1.81) |
| FIX | −0.000 | 0.000 | −0.000 |
| | (−1.03) | (0.90) | (−0.49) |
| ROA | 0.001 *** | 0.002 *** | 0.001 *** |
| | (3.73) | (4.32) | (4.43) |
| LEV | −0.110 *** | −0.078 *** | −0.069 *** |
| | (−4.42) | (−3.33) | (−3.55) |
| QUICK | 0.010 *** | 0.009 *** | 0.008 *** |
| | (2.87) | (2.87) | (2.79) |
| CSRR | −0.004 | −0.005 | −0.002 |
| | (−0.71) | (−0.95) | (−0.56) |
| OS | −0.020 *** | −0.014 ** | −0.009 * |
| | (−2.90) | (−2.27) | (−1.77) |
| OC | −0.004 | 0.011 | 0.008 |
| | (−0.26). | (0.66) | (0.62) |
| Constant | 0.115 ** | 0.073 * | 0.109 *** |
| | (2.46) | (1.69) | (3.00) |
| YEAR | YES | YES | YES |
| INDUSTRY | YES | YES | YES |
| Observations | 786 | 786 | 786 |
| R-squared | 0.183 | 0.294 | 0.481 |

*t*-statistics or z-statistics in parentheses. *** $p < 0.01$, ** $p < 0.05$, * $p < 0.1$.

## 5. Conclusions

Based on the environmental performance disclosed in the enterprise annual report, ESG report, corporate social responsibility report, and sustainable report, the listed companies in China's heavy pollution industries from 2010 to 2021 are taken as the research samples to explore the relationship between corporate sustainable development, environmental performance and financing costs in this paper. The research measures the

environmental performance of enterprises by the enterprise pollution charges and calculates the financing cost of enterprises by the ratio of enterprise interest expenditure to the average debt. Through the Breusch and Pagan-Lagrange multiplier test and Hausman test of random effects, it was determined that the use of fixed effect (within) regression or random effect GLS regression was the appropriate estimation method. The empirical results show that the better the environmental performance, the lower the cost of debt. Moreover, sustainable development weakens the inhibitory effect of environmental performance on the cost of debt. The stronger the sustainable development (internal growth capacity), the lower the debt cost. Finally, the following conclusions are drawn: the improvement of environmental performance will reduce the cost of debt financing. Higher enterprise environmental performance can improve enterprise value, reduce creditors' judgment on debt repayment risk, and reduce their required rate of return, thus reducing the cost of debt financing capital of enterprises. This also verifies the research results of previous scholars [45]. The improvement of environmental performance weakens the inhibition of sustainable development capability on debt financing cost. A higher level of enterprise environmental performance will send a positive signal, help enterprises obtain long-term and stable financial support in the period of seeking sustainable development, promote enterprises to achieve long-term, healthy and stable development, and then reduce financing costs.

This study contributed to the literature. First of all, good environmental performance can reduce the debt cost of enterprises, indicating that environmental performance has economic value. The study also shows that when assessing corporate risk, Chinese lenders take the environmental risk and environmental performance into consideration. Therefore, from the enterprise level, enterprises can learn from this study that they should actively focus on and improve environmental performance, and use this strategy to reduce debt costs. From the government level, relevant departments should establish reasonable environmental performance measurement and standards, and provide normative guidance for enterprises, which can effectively impose external restrictions on enterprises and prompt enterprises to improve their environmental performance. In addition, enterprises should also improve their sustainable development ability, so as to reduce debt costs and achieve long-term development of enterprises.

The level of enterprise environmental performance is a comprehensive concept, and the pollution charge is the cost for enterprises to bear their pollution emissions. Although this data has been audited by a third party, it cannot guarantee that the environmental performance can be accurately measured. Different measurement methods may lead to different research results. In addition, this study does not consider that the impact of enterprise environmental performance on financial performance is not only limited to debt financing costs but will also have an impact on financial indicators such as equity financing costs. Since debt financing is still the main way for Chinese enterprises to obtain external financing, debt financing costs are selected as the explanatory variable of this paper, without considering the impact on other statement items.

To sum up, in further in-depth research, we need to increase the consideration of variables such as capital structure and capital market development, eliminate relevant interference factors, and clearly show the relationship between sustainable development, environmental performance, and financing costs. The variable explained in this paper is debt financing cost, but the impact on other report items of the enterprise is also worth studying. Therefore, it is necessary to extensively study the changes of each item and comprehensively consider the real and comprehensive impact of the enterprise. For enterprise environmental performance, with the optimization of the green regulatory system and the continuous improvement of the environmental information database, a more scientific and reasonable environmental performance evaluation system should be built to comprehensively analyze and evaluate the level of enterprise environmental performance. Moreover, on the basis of available data, efforts should be made to explore whether the

relationship between sustainable development, environmental performance, and financing costs is different among different industries.

**Author Contributions:** Conceptualization, H.S. and G.W.; methodology, J.S. and J.B.; software, E.D.; validation, X.Z.; formal analysis, X.Z.; investigation, F.C.; resources, X.Z.; data curation, L.Z. and F.C.; writing—original draft preparation, E.D. and X.Z.; writing—review and editing, J.S.; visualization, X.Z.; supervision, H.S., G.W. and J.B.; project administration, J.S.; funding acquisition, J.S. All authors have read and agreed to the published version of the manuscript.

**Funding:** This research received no external funding.

**Institutional Review Board Statement:** Not applicable.

**Informed Consent Statement:** Not applicable.

**Conflicts of Interest:** The authors declare no conflict of interest.

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
