# Peer review of "Corporate Sustainable Development, Corporate Environmental Performance and Cost of Debt"

_sustainability, doi:10.3390/su15010228_

Round 1

Reviewer 1 Report

Thank you for the opportunity to review this manuscript. As a researcher in this field myself, I very much enjoyed reading it. This paper examines determinants of the cost of capital with focus on the corporate environmental performance and corporate sustainable development Chinese listed companies. While the article is informative and of potential interest among some readers, I found it difficult to justify the rationale and motivation for conducting the research. Below I describe my concerns in greater detail. I hope that my comments are viewed as constructive feedback that will help you clarify and enhance your research quality. The comments are as follows:

Major comments; lack of strong justification of the research. Although the authors mentioned that “the global environmental situation shows that the problem of environ[1]mental degradation is becoming more and more serious.” Of course, such statements indicates the importance of corporate sustainability and environmental aspects, but I believe that this reason alone is not enough to necessitate the research and there is no justification for its link to the sustainability and cost of capital. The author may need to discuss the rationale of the study in a critical manner link it to the other constructs included in this research.

Title;- sound!

Keywords;- add “costs of capital” “sustainability” “China”; these are just suggestion.

Abstract;-

-        All of the following statements to highlight the research objective “This paper examines the impact of corporate environmental performance and corporate sustainable development on financing cost in China. High corporate environmental performance may reduce the financing cost. Good environmental performance promotes enterprises to fulfill their social responsibilities, save energy and reduce consumption, create a good social image, and achieve sustainable development. Therefore, how to promote enterprises to improve environmental performance and actively fulfill their social responsibilities is the key to the further development of enterprises. The purpose of this paper is to discuss the mechanism of corporate environmental performance on financing cost, and whether the corporate sustainable development plays a regulatory role in the study of heavily polluted industries.” I suggest the authors to replace with single clear research objective. By doing so, the authors would have space to discuss the research findings further.

-        The authors need to further discuss the research findings.

-        Highlight the research contributions.

Introduction;- Apart from the major concern discussed above, the authors need to consider the followings;

-        References are missing from some important argument, not only in the introduction but throughout the manuscript. For example;” Brundtland first proposed the concept of "sustainable development", which is de[1]fined as "development that not only meets the needs of contemporary people, but also does not harm the ability of future generations to meet their needs"”.

-        The authors need to discuss the research problem in the introduction. As cost of capital is the main construct of the study, the author need to highlight with is the retionality about this topic that such study would help to solve.

-        Background about the context of the study (China) related to the research topic is needed.

-        In separate paragraphs, the authors need to explain how the study contribute/ add to the existing knowledge.

-        After the research background/retionality, research introduction should contain clear research objective which developed from the earlier sections

-        Is figure 1 necessary !

Literature review;-

-in section 2, the author should present theoretical ground for the research model.

-Update the reference sources used in this section.

- “Since the 21st century, in order to effectively reduce the company's financing costs, an overall and systematic review has been conducted from the inside ….” What do you mean?

-the author need to choose one hypothesis (positive/negative) and provide support to the chosen one. It is better not use contradict hypotheses. “H1a: Corporate environmental performance lowers the cost of debt financing. H1b: Corporate environmental performance increases the cost of debt financing.”

-argumentative discussion is needed.

Results;

-        I am not impressed with the discussion of the research result. The authors did not critical justify the analysis outcome or support/compare the result with prior studies findings.

-        The result is merely descriptive and not argumentative discussion was provided.

-        The authors used OLS regression or logit regression ?? “We employ logit models to model dichotomous dependent variables. The logit model with the better performance than the Probit model since it more appropriates for thought of as modeling log odds” “provides the OLS regression results of the relationship between sustainable development of enterprises,”

'

Conclusion;

-        the paper doesn't identify clearly any implications for research, it doesn't put forward a feasible research direction. The authors may need to elaborate how the study have valuable implications for policy maker and researchers.

-        Replace “5. Discussion and Conclusion” with “5. Conclusion”

There are a few of typos/grammatically incorrect sentences in the manuscript that need to be remedied. Please note that I provided just single examples below. The paper needs to be checked thoroughly to meet the standard required in this journal.
1     
There is a lack of smooth transition from one paragraph to another.

2      Kindly add the contribution of the authors paragraph after the conclusion, I would like to read the contribution of all the co-authors to this manuscript.

3      capitalization issues; see the headings as example; “b. Corporate sustainability and financing cost”….” 5. Discussion and Conclusion”, “4.4. Robustness Checks”…etc.

3     In-text reference and reference list is not as per the journal requirements.

4     write the equation of “c???????? ??????????? ??????????? ?????????y” in shourter and better form. Use abbreviations.

5    punctuation issues; “calculating the net profit growth level of enterprises in recent three years (Dhar et al., 2021)” this paragraph missing period (.).

6     Abbreviation should be defined first time used and consistently use it. Also, remove unnecessary abbreviations; ‘sustainable development (SUS)’, ‘enterprise financing cost (COST)’…. Etc.

7     As school of thought, I personally think that authors should avoid the use of personal pronouns (I, he, she ….) within the structured abstract and body of the paper (e.g., “He proposed and confirmed the conclusion that information asymmetry between fund”). Therefore, research writing should not be personalized. Rather it should be written from the perspective of a third party. Hence, if you agree please recheck the style of writing throughout the paper (wherever this applies).

consider seeing the following studies "A bibliometric analysis of cash holdings literature: current status, development, and agenda for future research" "Capital structure decisions and environmental, social and governance performance: insights from Jordan"

I believe your paper has merit and I am certain that it can be significantly improved. Once these comments addressed the paper will make an important contribution to the literature on this subject, Overall, I am delighted to read this manuscript. I hope that my comments and questions will provide the authors with some guidance to improve their manuscript.

Best of luck to you!

Reviewer 2 Report

The paper deals with a relevant and topical issue, i.e. measuring the impact of environmental and sustainable development on the cost of company financing. However, although the topic is relevant and the development of the methodology is adequate, the paper has some weaknesses that, in my opinion and unfortunately, do not allow me to recommend it for publication in the journal. These are:

- The literature review is not exhaustive and is not properly focused on the topic it addresses. The introduction and theoretical framework.

- The choice of variables is not adequately justified, especially in the cases of the independent variables, environmental development and sustainable development. They should be based on a thorough review of the literature.

- The results are not discussed in terms of the literature review. As a result, even if the discussion is timely and accurate, it loses value.

I encourage the authors to revise it thoroughly to address these problems.

Reviewer 3 Report

Dear authors,

This paper deals with an important and up to date topic which is the impact of  corporate environmental performance and sustainable development on financing cost  in the case of Chinese firms. The empirical results show that corporate environmental performance and sustainable development are negatively related with financing cost. The sustainable development  weakens the inhibition of corporate environmental performance on financing costs. 

the paper is well structured and the methodology is clear. however, the authors should take  into account the below remarks to further enhance the quality of the paper.

- First in the abstract, the authors wrote that " corporate environmental performance and sustainable development are negatively related with financing cost". the sentence is confusing , is there any moderator variable? if not please precise and make the sentence more clear. 

- in the introduction, line 93, the authors wrote :" few literatures discusses the relationship between sustainable development and financing costs (Liu et al., 2022" please correct it as  few literature discussed the relationship between sustainable development and financing costs (Liu et al., 2022)"

- figure 1 should be placed in the literature review

- in the literature review, the first hypothesis should be corrected. the authors should state only one hypothesis either Corporate environmental performance lowers or increases the cost of debt financing, based on the literature review. the same remark applies from hypotheses 2 and 3.

- in the methodology, the authors need to further justify the choice of the dependent variable by referring to the previous literature. i suggest the following recent article: W.Ghardallou (2022). Capital structure decisions and corporate performance: Does Firm’s profitability matter? Journal of Scientific & Industrial Research, 81 (8), pp. 859-865.

- in the results analysis, the authors can further justify the negative elationship between both main constructs by stating that increasing corporate social reponsibility enhances firm performance and thus reduces the cost of debts.  i suggest to include the following articles to justify this idea:

-W.Ghardallou, N. Alessa (2022). Corporate Social Responsibility and Firm Performance in GCC Countries: A Panel Smooth Transition Regression Model. Sustainability, 14(13): 7908.

- The conclusion and the policy implications are clearly stated and formulated.

Round 2

Reviewer 1 Report

Thank you for giving me the opportunity to review this manuscript again. I would like to congratulate the authors for all adjustments made in this paper, which definitely brought the paper to an upper level.

Best of luck!